# Inflammatory and Redox Mediators in Rat and Human Ovulation

**DOI:** 10.3390/ijms262411979

**Published:** 2025-12-12

**Authors:** Dorottya Varga, Péter Szatmári, Eszter Ducza

**Affiliations:** Department of Pharmacodynamics and Biopharmacy, University of Szeged, 6720 Szeged, Hungary; vdorottya0731@gmail.com (D.V.); szapeti40@gmail.com (P.S.)

**Keywords:** ovulation, inflammation, redox signaling, antioxidants, non-steroidal anti-inflammatory drugs

## Abstract

Ovulation is a critical event in mammalian reproduction, a complex process that involves the release of a mature oocyte from the ovaries for fertilization. Hormonal shifts are the driving force of the ovulation cycle; however, several other factors are able to fine-tune the occurrence of follicular rupture. Prior to the follicular rupture, the pre-ovulatory luteinizing hormone (LH) surge triggers a self-generating local inflammatory and redox cascade, which is responsible for the release of several inflammatory and redox signaling mediators. Eicosanoids are one of the key regulators of the initiation of the local inflammation within the follicle, while the balance of reactive oxygen species and antioxidants is fundamental to maintaining the physiologically coordinated redox state during the ovulation process. In this review, we aim to provide a summary of the human menstrual and rat estrus cycles and demonstrate the LH-induced inflammatory and redox cascade involved in follicle rupture through the details of lipid-derived and redox signaling mediators.

## 1. Introduction

Knowledge of the ovulation process is crucial in both human and animal models for several reasons. Knowledge of the process and timing of ovulation is essential for determining the fertility window (use of contraceptives), exploring causes of infertility (e.g., polycystic ovary syndrome, PCOS), planning in vitro fertilization (IVF), and other assisted reproductive techniques. However, accurate determination of the time of ovulation is also important in animal breeding, as accurate knowledge is required for timely, successful reproduction. Due to the ethical difficulties of human studies, choosing an appropriate animal model for studying ovulation is very important.

Inflammation and redox reactions have a double-edged sword nature on the reproductive system, since both of them have positive and negative impacts on fertility performance. The direction of these effects depends on the duration and extension of these processes. Under physiological conditions, ovulation is characterized by moderate inflammation and redox reactions as well. The ovulatory inflammatory event is a well-coordinated, time-controlled, low-grade inflammation localized to the follicle. In contrast to this, ovulatory redox molecules are generated continuously, and antioxidants balance their levels and actions, reducing the cellular damage. If these processes become uncontrolled, the local low-grade inflammation could turn to prolonged and generalized, while the rate of free radicals exceeds the antioxidant capacity, resulting in oxidative stress. The overproduction of inflammatory and redox signaling molecules in the ovary could induce tissue degradation and dysfunction, which contribute to the manifestation of infertility. Nevertheless, insufficient inflammatory and redox signaling during ovulation can also impair the reproductive outcomes.

Excessive exposure to antioxidants and nonsteroidal anti-inflammatory drugs (NSAIDs) can interfere with the ovulatory inflammatory process. The use of antioxidants, e.g., nutritional supplements, vitamins, and functional foods, is extremely popular; people see them as the key to a healthy lifestyle, which is partly true. NSAIDs, as over-the-counter drugs, are widely available and often used medicines to relieve pain and reduce inflammation worldwide. 

The primary purpose of our review is to summarize and integrate existing information on the human and rat ovarian cycles to provide a comparative perspective on the ovulation process between the two species and to highlight the potential to modulate ovulation via inflammatory and redox signaling cascades. Secondly, this review summarizes the effects of excessive exposure to antioxidants and NSAIDs on ovulation capacity.

## 2. Literature Search

We conducted a narrative review to synthesize the existing literature on the rat and human ovulatory processes involving ovulatory inflammatory and redox mediators. This method provides a flexible and comprehensive approach to summarizing and comparing diverse studies and to identifying underlying mechanisms and relationships. The review follows a structured narrative format with literature search, screening, and synthesis procedures. 

A comprehensive literature search was performed in four electronic databases, including PubMed, Scopus, Web of Science (WoS), and Google Scholar, for studies with titles or abstracts containing the keywords used to construct the searches. The following keywords were used but not limited to, “ovulation”, “ovary”, “follicle”, “fertility”, “hormonal levels”, “menstrual cycle”, “estrus cycle”, “human”, “rat”, “inflammation”, “redox signaling”, “cascade”, “oxidative stress”, “rupture”, “mediators”, “eicosanoids”, “prostaglandins”, “leukotrienes”, “reactive oxygen species”, “antioxidants”, “non-steroidal anti-inflammatory drugs”. These terms were used alone or in combinations using the Boolean operators “AND” and “OR”. The search was limited to publications in the English language. Neither publication type nor time restrictions were applied. The identified articles in the literature search were screened and selected by the authors based on the review of the title, abstract, and text. Those studies were integrated in the narrative summary, which ascertained their suitability and contribution to the objective of the review. To provide broader coverage of relevant sources, a forward-and-backward citation-chaining search methodology was also applied, using the references cited in the articles identified in the initial search. 

All information was organized and structured thematically, and references were managed by Zotero software (version 7.0.24). At the beginning of each section, essential information about the given topic was provided, then the underlying mechanisms, functions, controversies, clinical aspects, and future research directions were detailed according to the existing literature data.

## 3. Ovarian Activity

### 3.1. Oocyte Release

In the ovary, each oocyte is covered by a single granulosa cell layer, which is termed a primordial follicle. The initial oocyte pool at birth is approximately 1–2 million and decreases to around 300,000 at puberty. The primordial follicles remain quiescent until sexual maturation, then are continuously activated in a limited number by the regulation of the ovarian microenvironment and enter the folliculogenesis [1,2,3,4]. Once activated, they begin a slow maturation process with the contribution of intra- and extra-oocyte regulatory factors. First, the surrounding granulosa cells undergo a cuboid morphological change, resulting in primary follicles. During the second stage of the maturation process, follicles develop into secondary follicles by growing multiple granulosa layers surrounding the oocyte. With the appearance of the outer layer of the cells called theca, the follicles are encapsulated and form the preantral follicles. As the preantral follicles’ size increases, several fluid-filled cavities appear and merge into an antral cavity, thereby reaching the tertiary or antral stage of the follicles. These series are under paracrine control and gonadotropin independence; however, gonadotropin hormone is required for the further development of the antral follicles [3,5,6]. Antral follicular size is associated with higher follicle-stimulating hormone (FSH) receptor expression, which supports the larger follicles to become the dominant mature follicles and promotes the smaller follicles to undergo atresia [5,7]. The dominant follicles rapidly proliferate and express a high amount of luteinizing hormone (LH) receptors under the influence of FSH and transform into a pre-ovulatory/Graafian follicle. Since the Graafian follicles are sensitive to the LH fluctuations, the increased LH level is able to induce follicular rupture and the release of the mature oocyte for fertilization. This event is called ovulation, which is repeated periodically and depends on the hormonal cycle [5]. The number of ova released at ovulation and the length of the hormonal cycle vary between mammals.

### 3.2. Ovarian Cycle in Human and Rat

The menstrual cycle is under the coordination of the mature hypothalamic-pituitary-ovarian (HPO) axis. Puberty provokes a rise in amplitude and frequency of the pulsatile release of gonadotropin-releasing hormone (GnRH) in the hypothalamus, promoting the secretion of FSH and LH in the pituitary gland, which stimulates the development of the follicles in the ovaries and initiates the reproductive cycle [8,9,10]. The average menstrual cycle covers a 28-day period; however, it occurs at intervals of 21 to 35 days. Ovulation halves the ovarian cycle, dividing it into follicular and luteal phases; however, according to the hormonal changes, the two main phases can be further divided into subphases as early and late follicular and early, mid, and late luteal phases [11,12]. During the menstrual cycle, the GnRH pulses determine the hormonal levels by positive and negative feedback regulatory mechanisms, which become more frequent in the follicular phase and ovulation, then decrease in the luteal phase [13]. The hormonal changes in each menstrual phase are represented in Figure 1.

The follicular phase corresponds with the follicular recruitment and maturation process, which begins in each cycle and lasts for 14 days, until ovulation occurs [14]. At the early follicular phase between days 1–5, menstruation occurs with low and stable sex hormone levels [12,13,15,16]. Following menstruation, the level of FSH slightly rises and stimulates the maturation of the gonadotropin hormone-dependent antral follicles. During the late follicular phase between 6–14 days, the growing follicles are progressively producing estrogen and inhibin B by their granulosa cells, providing negative feedback on the HPO axis to suppress the FSH secretion and restrict the smaller follicles’ development. As a result, a dominant follicle emerges and transforms into a Graafian follicle, while the non-dominant follicles degrade [5,17]. At the end of the follicular phase, the estrogen level rises exponentially by the dominant follicle, reaching a critical threshold that results in positive feedback on the pituitary gland, sensitizing the gonadotrophic cells by the increased expression of GnRH receptors [18]. As a consequence, a sudden increase in LH secretion occurs with a low increase of FSH and finishes the follicular phase and initiates the ovulation process at days 13–15, 12–36 h later, the LH surge [17,19,20,21]. The level of prolactin remains at low basal levels through the follicular phase, then slightly increases at the time of ovulation [15,16].

The pre-ovulatory LH surge induces follicle rupture and releases the oocyte from the ovary. After the ovulation, the luteal phase ensues, which lasts for 14 days, until the start of the next cycle. At the early luteal phase between 16–20 days, the ruptured follicle transforms into the temporary corpus luteum by the influence of LH and starts to secrete progesterone, moderate estrogen, and inhibin A [12,13]. Subsequently, progesterone reaches a peak concentration, and a second small estrogen peak is formed. During the mid-luteal phase from 21–23 days, the elevated levels of progesterone, estrogen, and inhibin A induce negative feedback on the pituitary gland, resulting in low levels of FSH and LH, preventing the maturation of new follicles in the luteal phase. In the late luteal phase, around day 24–28, the corpus luteum breaks down during luteolysis and transforms into the non-functional corpus albicans. Consequently, the progesterone and estrogen secretion stop and sharply decline with the level of inhibin A as well [19]. By this, FSH is able to rise again in the last few days of the luteal phase while the prolactin is returning to baseline levels, then the menstrual cycle restarts with the next follicular stage [11,12,15,17].

During the menstrual cycle, the endometrium also responds to the hormonal alteration. The increasing estrogen level during the follicular phase stimulates the development of the endometrium thickness and increases the depth of the spiral arteries. Without fertilization, the level of progesterone and estrogen declines, and the developed endometrium lining sheds with bleeding [22]. If implantation occurs, the fertilized egg attaches and invades the endometrium and starts to secrete human chorionic gonadotropin (hCG) hormone, which maintains the corpus luteum and the high levels of progesterone and estrogen. As a result of this, the endometrial stromal cells transform into decidual cells to accommodate pregnancy [23]. In humans, typically one oocyte is released from each ovary in each cycle; however, it is rare for more than one oocyte to be released, which can result in dizygotic twins [24].

The estrus cycle is the main reproductive cycle of non-primate mammals, including rats. Female rats experience their first estrus event early after birth, around the age of four weeks, as puberty occurs; however, the entire sexual maturity with regulated cycles is dated around the age of 6–8 weeks [25,26,27]. The average cycle of female rats repeats every 4–5 days, which is divided into four stages: proestrus, estrus, metestrus, and diestrus. The HPO axis regulates these stages, and each is characterized by specific cell types in vaginal smears and hormonal shifts [28]. In addition, the circadian rhythm, with synchronized light-dark periods, also influences the cycle [29]. The hormonal changes in each estrus phase are represented in Figure 2.

The estrus cycle begins with the proestrus phase, which lasts 12–14 h. During this time, follicles go through the same maturation stages as humans, showing similar gonadotropin dependence in the antral stages. As the dominant follicles are selected, estrogen is secreted, reaching a pre-ovulatory peak which contributes to the rise of LH and FSH concentrations. At the same time, progesterone and prolactin levels also rise, creating similar pre-ovulatory peaks that coincide with the LH surge [15,30,31,32,33]. The sharp LH peak, with the drop of estrogen and progesterone, triggers ovulation at the beginning of the estrus phase. Ovulation happens 10–12 h after the LH peak. The inhibin A level is higher before ovulation than after, while B falls below its baseline level around the period of ovulation [34]. The length of the estrus phase is 24–48 h when the ruptured Graafian follicles start to transform into corpora lutea, accompanied by a decrease in estrogen, LH, FSH, and prolactin levels [11]. The newly formed corpora l-tea show high steroidogenic and low luteolytic gene expressions, thereby secreting progesterone for a short period from nearly the end of estrus and during the 6–8 h lasting metestrus phase [35,36]. After the short metestrus, the luteolytic genes become predominant, and the corpora lutea begin to regress through luteolysis in the following diestrus phase, which lasts 48–72 h [11,37,38]. The progesterone level started to decrease as the corpora lutea regressed in the absence of prolactin [39]. At the end of di-estrus, the cycle starts over; however, the luteal regression process of copora lutea is prolonged over several estrus cycles [11,36,40].

Rats are sexually receptive at the phase of estrus and are characterized by bilateral ovulation; therefore, they release multiple oocytes from both ovaries at the same time within a cycle, resulting in larger litter sizes. If conception does not occur, after an initial thickening, the endometrium reabsorbs in the diestrus phase in each cycle [41]. In case of fertilization, the pituitary gland starts to secrete prolactin at the beginning of pregnancy to prevent the corpus luteum breakdown and provide progesterone production for the gestation period [26,39,42,43].

Several similarities and differences could be observed between the rat and human cycles. The main difference is the cycle length, since the rat estrus cycle is relatively short in contrast to the human. As a result, the division of the cycles also differs. In humans, ovulation is pericentral, halves the 28-day menstrual cycle, and occurs around day 14 of the 28-day cycle; however, each phase has its own counterpart. The human follicular stage corresponds with the proestrus phase, with a similar follicular development process, while the phases of metestrus and diestrus show the characteristics of the luteal phase. A significant difference is that in humans, the corpus luteum functions until the end of the luteinizing phase, then degrades, while in rats, it regresses after a short period of function, which lasts for several cycles. In each cycle, several eggs mature at the same time in rats, while in humans, usually only one does. Moreover, the endometrium lining is shed with bleeding in humans in the absence of fertilization, while in rats, it is reabsorbed without bleeding. The hormonal profiles follow similar patterns, with minor differences between the two species. The unique characteristics of the two cycles are summarized in Table 1.

## 4. Inflammation and Redox States in Ovulation

### 4.1. LH-Induced Inflammatory and Redox Cascade

Most of the processes involved in follicle rupture are coordinated by inflammation and oxidative events upon LH stimulation. Granulosa and theca cells express abundant LH receptors; therefore, they are the first to respond to the LH stimulus. By the stimuli, they release several proteolytic enzymes, including matrix metallopeptidases (MMPs), which induce the fragmentation of the collagenous matrix and disrupt the basal lamina of the extracellular matrix (ECM). As the basal lamina degrades, new vessels grow into the granulosa compartment. In parallel, granulosa and theca cells started to form site-acting agents like eicosanoids and reactive oxygen species; moreover, they secrete cytokines (interleukins (IL, including IL-1β, IL-6), tumor necrosis factor-alfa (TNF-α) and chemokines, which initiate the infiltration of the circulating leukocytes into the follicle [44,45,46,47]. The infiltrated leukocytes also secrete cytokines, chemokines, and proteases; moreover, they also produce eicosanoids and free radicals to further induce the progression of inflammation and ECM degradation. The cumulus granulosa cells also secrete these inflammatory and oxidative mediators, which induce cumulus expansion and contribute to the detachment of the cumulus oocyte complex (COC) from the basal granulosa cells [47,48]. The loss of the granulosa and theca cells at the apical site of the follicle opens the wall and leads to the follicle rupture and the release of the oocyte [44,47]. These processes induce a self-generating inflammatory cascade reaction with several positive feedback loops, which amplify the LH stimulus and lead to a local and acute inflammation in the ovary during ovulation (Figure 3).

As described, numerous autocrine and paracrine intrafollicular regulators are produced due to the LH surge, activating the diverse interrelated biochemical signaling cascades. Since ovulation is an inflammatory and an oxidative event as well, in the following chapters, we summarize the role of two locally produced and acting mediators in ovulation, including lipid-derived and redox signaling mediators. Both lipid-derived and redox signaling molecules are active components of the integrated ovulatory signaling network, which is responsible for the initiation and amplification of the ovulatory inflammatory and redox cascade. It is important to mention that during the self-generating cascade reactions, these molecules interact, amplify, and regulate each other’s productions and actions simultaneously to drive the ovulatory response. Since these cascade reaction processes are very complex and multifactorial, we detail the selected mediators separately for better understanding.

### 4.2. Lipid-Derived Mediators in the Follicle

Eicosanoids are the primary factors in initiating the inflammatory cascade. They are bioactive lipid molecules and are not stored by the cell; therefore, they typically exert their effects in an autocrine or paracrine manner near their site of synthesis [49,50]. The main precursor of the eicosanoids is the arachidonic acid (AA), which is cleaved from the phospholipid membrane by phospholipase A2 (PLA2) subtypes. Following cleavage, three pathways become available for the AA conversion. One way is the cyclooxygenase pathway, which produces the short-lived intermediary prostaglandin H2 (PGH2), and then further converts it to different types of prostaglandins by prostaglandin synthase (PGES) or to thromboxane by thromboxane synthase. Another way of the conversion is the lipoxygenase pathway, which results in leukotriene production [49,50,51]. Eicosanoids have a short lifespan, but they are involved in numerous physiological processes, including the regulation of vascular permeability, platelet activation, inflammation, chemotaxis, and immune response [49]. All of these types of eicosanoids are identified in the ovulatory follicles with different roles and importance during ovulation.

#### 4.2.1. Prostaglandins

The role of prostaglandins in ovulation induction is widely studied and established. Their key regulator role in the induction of ovulation is proven. Currently, four types of prostaglandins exist, including prostaglandin E2 (PGE2), prostaglandin D2 (PGD2), prostaglandin F2-alfa (PGF2α), and prostaglandin I2 (PGI2). In the follicle, granulosa and theca cells express the key enzymes of these prostaglandins as prostaglandin synthase protein 1 (PTGS1) or cyclooxygenase-1 (COX1) and prostaglandin synthase protein 2 (PTGS2) or cyclooxygenase-2 (COX2), which produce PGH2. In addition to the conversion of PGH2 by PGESs, aldo-keto reductase isoenzymes (AKR, including AKR1C1, 1C2, and 1C3) also convert PGH2 or PGD2 to PGF2α, and interconvert PGE2 and PGF2α. Prostaglandins act through prostaglandin receptors, which have several types and each shows varied affinity, signaling pathways, and regional expression within the follicle, explaining the diverse functions of the prostaglandins [52,53,54,55]. In the ovulatory follicle, PGE2 is the pivotal prostaglandin, followed by PGF2α, and they are the essential paracrine mediators of the LH surge and show positive correlation with each other in the follicular fluid during the ovulation cascade. Prostaglandin E2 has four types of receptors (EP), as EP1-EP4, while PGF2α has one receptor, formerly FP. All of these receptors are coupled with varied guanine-nucleotide-binding (G) proteins, resulting in the different intracellular responses [46,56]. Numerous studies have examined the potential role of prostaglandins in the ovulation process in several species, and most cases highlight that a decreased level of prostaglandins, altered receptor expression, or function is associated with abnormal COC expansion, decreased oocyte maturation, and unruptured follicles [52,55].

Up to the LH surge, the LH receptor-rich granulosa and theca cells increase the level of prostaglandins in the intrafollicular compartment, though in several ways. The LH stimulus increases the expression of enzymes involved in the synthesis or conversion of prostaglandins, such as COXs, prostaglandin E synthases (converts PGH2 to PGF2α), and aldo-keto reductases (converts PGH2 to PGF2α and PGE2 to PGF2α) [46,51]. LH also induces transporter elevations, which are involved in their distribution within the follicle. LH-triggered increased solute carrier A1 (SLC2A1) supports the passage of prostaglandins into the cell, while ATP-binding cassette C4 (ABCC4) provides movement out of the cell into the follicular fluid. In addition, LH is able to restrict the expression or activity of enzymes responsible for prostaglandin metabolism, such as prostaglandin dehydrogenase. All of these mechanisms contribute to the increased level of prostaglandins within the follicle. Moreover, LH increases the expression of various prostaglandin receptors, thereby enhancing the prostaglandin-induced response [51].

Prostaglandins can act on the follicular vasculature as both vasoconstrictors and vasodilators. At the apical site of the follicle, they induce vasoconstriction, which supports the rupture of the wall, while in the rest of the follicle, vasodilatation is the dominant response to provide increased vessel permeability and blood flow for angiogenesis and the leukocyte infiltration into the follicle [46]. The follicle wall contains smooth muscle cells, which are sensitive to prostaglandins. PGE2 induced smooth muscle contraction in these cells, which suggested contributing to the apical thinning of the wall [57]. Prostaglandins also regulate the cumulus expansion and oocyte maturation processes. The increased expression of PGE2 upregulates several genes in the granulosa and cumulus cells, which are involved in the cumulus expansion process, including amphiregulin (*Areg*), epiregulin (*Ereg*), hyaluronan synthase 2 (*Has2*), and TNFα-induced protein 6 (*Tnfαip6*). In addition, PGE2 also induces cyclic adenosine monophosphate (cAMP) elevation in the cumulus cells to increase the gene expression of *Has2* and *Tnfaip6*. PGE2 also activates the protein kinase B and mitogen-activated protein kinases3/1 (PKB-MAPK3/1) pathway, which regulates the cumulus expansion and oocyte maturation [55]. To release the oocyte from the follicle, extensive ECM remodeling and degradation are needed, which is regulated by diverse proteolytic systems. Beyond the wall rupture, remodeling is also important for the formation of the corpus luteum. Increased PGE2 levels mediate the plasminogen-dependent proteolysis in the periovulatory follicle [58], while PGF2α modifies the follicular collagen synthesis through activating collagenases to induce collagenolysis [59]. Moreover, prostaglandins also increase the messenger ribonucleic acid (mRNA) level of ADAMTS proteases (a disintegrin and metalloprotease with thrombospondin motifs) [60,61]. More studies revealed that PGE2 has a protective role during the luteolysis, while PGF2α has a positive feedback and an initiation factor to the induction of corpus luteum regression [62,63]. These effects are dependent on the receptor distributions induced by progesterone in the luteal cells. Moreover, an intraluteal positive feedback mechanism for PGF2α secretion has been revealed, known as the auto-amplification pathway, which increases the level of PGF2α in response to PGF2α [64]. These data indicate that prostaglandins mediate almost all of the vital processes necessary for follicle wall rupture and oocyte release.

In addition to the prostaglandins, the COX pathway also produces thromboxane (TX), which participates mainly in platelet aggregation. Few studies are available regarding its role in the development of ovarian follicles and growth rate. Thromboxane A2 (TXA2) and its metabolite thromboxane B2 (TXB2) are present in the follicular fluid by the secretion of granulosa cells [56]. TXB2 level is inversely correlated with the follicle size, and higher levels of TXB2 in the follicle fluid are associated with lower oocyte maturity [65,66,67]. The level of TXB2 is increasing before and remains elevated after ovulation; however, the TXB2 rise occurred much later than the prostaglandin elevation. As such, elevated expression of thromboxane may not be an obligatory component of the ovulatory mechanism; nevertheless, TXB2 elevation coincides with thecal aggregation of blood platelets, which may control the blood loss during the follicle rupture [59].

#### 4.2.2. Leukotrienes

Leukotrienes are also arachidonic acid-derived bioactive mediators that are formed by the 5-lipoxygenase (LOX). Firstly, LOX converts arachidonic acid to leukotriene A4 (LTA4), which is further hydrolyzed or conjugated with glutathione to form LTB4, LTC4, LTD4, or LTE4 types. These products from the LOX pathway are involved in the induction of follicle rupture, especially LTB4 [68]. All leukotrienes are expressed in the ovulatory follicle with similar concentrations as prostaglandins; however, no correlation between prostaglandin and leukotriene levels was observed in the follicular fluid [56]. Leukotriene levels are increased during ovulation as PGE2 and PG2α; however, the elevation time courses for each eicosanoid were different [69]. The concentration of leukotrienes also increased when the COX pathway was blocked by indomethacin, caused by the arachidonic acid shunt from the COX pathway to LOX. Leukotrienes show several inflammatory properties all over the body, including increased microvascular permeability, promoting chemotactic movement of leukocytes, and inducing proteolytic release from these leukocytes, and more of these functions are identified in the follicle also [70]. The increased level of leukotrienes, mainly LTB4, is responsible for the degradation of the collagenous extracellular matrix of the apex of the follicle by the modulation of MMP-2 [71]. It is also proven that LTB4 has luteoprotective properties, while LTC4 has luteolytic properties. LTB4 increases, since LTC4 decreases the secretion of progesterone, which influences the lifespan of the corpus luteum. However, LTB4 presumably is not involved in the regulation of meiotic maturation [72,73]. According to the facts mentioned above, leukotrienes have similar ovulatory properties as prostaglandins; however, their role is mainly limited to the support of leukocyte infiltration into the follicle.

### 4.3. Redox Signaling Mediators in the Follicle

#### 4.3.1. Reactive Oxygen Species

Reactive oxygen species (ROS) are free radicals and byproducts of cell metabolism. These active oxidation substances are highly reactive, short-lived, and unstable; moreover, similarly to prostaglandins, they act locally on their synthesis site [74]. Reactive oxygen species include the superoxide anion (O_2_^−^), hydroxyl radical (OH^−^), and hydrogen peroxide (H_2_O_2_); however, nitrogen-containing reactive species also exist as nitric oxide (NO^−^) and nitrogen dioxide (NO_2_) [75]. Oxyradicals are mostly generated in the mitochondria during the electron transport chain. The leaked electrons react with the molecular oxygen, which results in the production of ROS. Due to their high reactivity, they react quickly with their environment without selectivity after their formation; therefore, protein and deoxyribonucleic acid (DNA) damage or lipid peroxidation occur. Their effect on the cells can be dual, depending on their concentration. In physiologically controlled levels, they act as a second messenger in several physiological processes all over the body, including ovulatory follicles [74,75].

ROS are involved in both folliculogenesis and luteogenesis. As the follicle grows and the oocyte develops, rigorous energy supply is necessary, which is provided by the mitochondria. The increased adenosine triphosphate (ATP) generation during the maturation process results in increased production of ROS by the mitochondrial electron transport chain, which modulates the oocyte maturation and quality by meiotic resumption. The generated ROS also triggers the follicular atresia by inducing granulosa cell apoptosis in the follicular stage. Furthermore, in the luteal phase, ROS induces LH sensitivity loss and steroidogenic properties in granulosa cells, contributing to the regression of the corpus luteum [75,76]. Nevertheless, excessive ROS generation—caused by dysfunctional mitochondria or due to mitochondrial DNA mutations or alterations in respiratory chain complexes—can manifest into oxidative stress and damage the cellular integrity [48,74,75,77]. To avoid this, varied enzymatic and non-enzymatic defense systems are developed to balance the ROS concentration by neutralization.

#### 4.3.2. Antioxidants

In the follicle, several enzymatic antioxidants are present [78,79,80,81,82], and the three most common enzymes are the superoxide dismutase (SOD), catalase (CAT), and glutathione peroxidase (GPx). SOD catalyses the dismutation of the free radical superoxide anion, which is the initial step of the ROS detoxification and results in molecular oxygen (O_2_) or hydrogen peroxide [83]. SOD can occur in three forms in mammals: copper-zinc-containing, cytosolic (SOD1), manganese-containing, mitochondrial (SOD2), and extracellular SOD (SOD3). The expression of types of SOD closely correlates with steroidogenesis in the human ovary [84]. SOD1 isotype is found in the cytoplasm of ovarian cells, such as granulosa and theca cells. Neither SOD1 nor SOD2 has been observed in primordial and primary follicles. SOD2 has been detected in secondary follicles, while SOD1 begins to appear in theca cells after the formation of the antral cavity. SOD1 cannot be detected in granulosa cells until follicles enter the dominant follicle stage. SOD1 null mice exhibit reduced fertility, as evidenced by a decreased number of pre-ovulatory follicles and corpora lutea, as well as primary and small antral follicles. In contrast to SOD1, SOD2 concentration in the corpus luteum is enhanced in the regression phase to clear the excess ROS produced in mitochondria by cytokines and inflammatory reactions. Thus, SOD1 activity in the corpus luteum is closely correlated with progesterone secretion, while SOD2 is primarily targeted to protect the luteal cells from oxidative damage caused by inflammation [83]. SOD3 is the only SOD isoform detectable in the zona pellucida. Interestingly, SOD3 can be translocated from cumulus cells into oocytes under certain conditions [84]. This evidence may demonstrate that SOD1 and SOD3 potentially contribute to the protection of DNA or transcription regulation of redox-sensitive genes. After SOD converts ROS to H_2_O_2_, catalase in the peroxisomes or GPx in the mitochondria completes the neutralization process and converts it to water (H_2_O) [83]. Peroxiredoxines (Prdx) are also enzymatic antioxidants that catalyze the conversion of ROS into H_2_O_2_ using their cysteine residue reactivity [85]. The Prdx family has several members, and Prdx2 and Prdx4 are found in the follicular compartment [86]. Prdx4 is expressed in the mouse COC and human follicular fluid and positively correlates with oocyte fertilization and the quality embryo rate [87]. As for Prdx2, it is revealed that it is expressed in rat antral and pre-ovulatory follicles and stimulated by gonadotropin surge [88]. In addition to enzymatic antioxidants, nuclear factor erythroid 2-related factor 2 (Nrf2) is a key regulator of oxidative homeostasis, playing a crucial role in defending against oxidative stress and enhancing ovarian function in women of reproductive age [89]. Sindan et al. found that the Nrf2 protein was mainly localized in the granulosa cells and oocytes of the secondary and antral follicles in mouse ovaries [90]. In addition, *Nrf2−/−* mice had fewer remaining primordial follicles at 10–12 months of age than wild mice [90,91]. This suggested that the antioxidant capacity of the ovary may decrease with age and that *Nrf2* knockout accelerates ovarian aging. Throughout the stages of youth, growth, and aging, as ovarian reproductive function initially rises, the levels of Nrf2 protein within the ovarian tissue demonstrate a pattern of increase followed by a subsequent decline.

Non-enzymatic antioxidants are endogenous or synthetic substances, minerals, or dietary constituents, such as glutathione, vitamin C, vitamin E, selenium, or zinc. Glutathione is an endogenous molecule that has reduced (GSH) and oxidized (GSSH) forms in the body. The reduced form of glutathione donates electrons to the free radicals and transforms into oxidized glutathione. Following the transformation, GSSH is converted back to GSH via the consumption of nicotinamide adenine dinucleotide phosphate (NADPH) by glutathione reductase, thereby restoring the antioxidant capacity of GSH. Vitamin C is one of the electron donors; it supports the GSH recycling with the reduction of free radicals [77,92,93,94].

In contrast to vitamin C, vitamin E (α-tocopherol) is an important lipid-soluble antioxidant found in the cell membrane that protects against lipid peroxidation by acting as a hydrogen donor to scavenge free radicals. It can positively modify oxidative stress biomarkers, improve erythropoiesis, and contribute to the stabilization of atherosclerotic plaques. Following the neutralization process, vitamin E becomes oxidized, and GSH and vitamin C support the transformation back to the active form [95,96]. Selenium and zinc act as antioxidants, but they cannot be considered antioxidants in their own right. Both minerals serve as cofactors of antioxidant enzymes SOD and GPx [22].

Numerous studies revealed the beneficial effects of antioxidants in oxidative stress on the follicular and oocyte development, including the reduction of DNA damage, increase of maturation rate, and the quality of oocyte, or the promotion of survival rate of pre-antral follicles from atresia. If excessive ROS cannot be balanced by antioxidants, the oocyte maturation and development could be damaged and lead to fertility impairments [97,98,99].

Following the LH surge, cells in the ovary produce ROS, which trigger essential events like cumulus expansion and follicle rupture, ultimately leading to egg release [94,100]. Nevertheless, few studies examined the direct effect of antioxidants on the ovulation rate. Miyazaki et al. examined the effect of SOD and its combination with CAT on ovulation in the rabbit ovary. SOD alone and combined with CAT suppresses the ROS production and reduces the ovulation rate without the induction of degeneration in the follicular oocytes [101]. In another study, Sato et al. injected long-acting SOD into rats intravenously, which also inhibited the ovulation [102]. Tamate et al. treated rats with the Copper (Cu), Zinc (Zn)-SOD, and Manganese (Mn)-SOD isoforms and found that both types of SOD reduced the number of ova compared to control animals [103]. Park Jl. et al. demonstrated the induction of H_2_O_2_ signaling and the antioxidant sulfiredoxin by an endocrine hormone (LH/hCG) in pre-ovulatory granulosa cells of the rat ovary. Administration of antioxidants in rats inhibited ovulation rate and cumulus expansion [88]. The LH-induced cumulus mucification/expansion by *Ptgs2*, *Has2*, and *Tnfaip6* upregulations was prevented by antioxidants both in vivo and in an ex vivo system of isolated intact ovarian follicles. Progesterone level also decreased in LH-induced ovulation in the presence of antioxidants. Along this line, H_2_O_2_ induced p42/MAPK signaling activation and fully mimicked the effect of LH, bringing about an extensive mucification/expansion of the follicle-enclosed COCs [104]. Taken together, these studies suggest that while antioxidants generally protect against cellular damage, they can interfere with ovulation by inhibiting the production of reactive oxygen species that are necessary for ovulation.

### 4.4. Aspects of Eicosanoids and Antioxidants in Human and Rat Ovulation

Several studies have examined the levels of eicosanoids and antioxidants in humans and rats (Table 2). In humans, most data are available from follicular fluid, while in rats, whole tissues are mainly studied, supplemented with in vivo or ex vivo pharmacological treatments. According to these studies, both humans and rats exhibit similar characteristics regarding the aforementioned mediators and antioxidants, which are influenced by the LH surge. Since in vivo human reproduction investigations are limited by certain ethical concerns [105], the rat model could be promising for investigating the inflammatory and oxidative pathways in ovulation.

## 5. NSAIDs and Their Influence on Ovulation

More studies have investigated the effects of pain-relieving medication use on conceiving and ovulation and have shown conflicting results. The role of COX activity is known in both inflammation and ovulatory events. One isoform of COX, PTGS1 (COX1), is present in the ovary during ovulation, but its levels do not correlate with elevated follicular prostaglandin levels. In contrast, PTGS2 (COX2) is responsible for peri-ovulatory prostaglandin production by ovarian follicles [52,53,54,55,127]. A genetic knockout of the *Ptgs2* gene in mice results in profound reproductive dysfunction, including anovulation and defective fertilization, underscoring the enzyme’s indispensable role in female reproductive physiology [128]. More animal and human studies have demonstrated that NSAIDs can influence the fertilization rates by the inhibition of COX enzymes with different selectivity [129] (Table 3).

NSAIDs, such as ibuprofen, are frequently used as over-the-counter analgesics by reproductive-aged women. In a study by Wolff et al., it was determined that ibuprofen should not be taken in a daily dosage of 400 mg three times a day around the time of ovulation in women trying to conceive, as this dose can delay ovulation. This ovulation-delaying effect of ibuprofen can be used in infertility treatments in which some delay of ovulation is required. This might include IVF treatments in which a beginning LH surge is detected, and ibuprofen is given to delay ovulation, to aspirate follicles 2 days later [130]. In contrast, Kohl Schwartz et al. found that oocyte maturation, embryo quality, and implantation rates were not negatively affected by ibuprofen (400 mg) intake [131]. Bou Nemer L et al. sought to evaluate whether a single oral dose of ibuprofen (800 mg) administered 15–18 h post-trigger injection would affect the follicular fluid (FF) levels of inflammatory markers involved in ovulation. Their data show that one dose of ibuprofen administered orally the day after trigger injection revealed a significant impact on the FF inflammatory milieu [132].

Naproxen is widely consumed by women because it is effective as an analgesic and anti-inflammatory; therefore, it is used to treat menstrual cramps. Its pharmacological effects are related to inhibiting COX, reducing the production of prostaglandins (PGF2α and PGE2). In the ovaries, PGE2 interacts with the PGE2-receptor to induce the expression of genes critical for ovulation. Administration of naproxen at 10 and 50 mg/kg for eight consecutive days altered murine follicular growth and ovulation. The effects of naproxen on fertility in female mice were observed in later classes of follicular growth, and these data, for the first time in the literature, showed downregulation of insulin-like growth factor 1 receptor (Igf1r) and reduced granulosa cell proliferation in 10 and 50 mg/kg naproxen-treated females [133]. McInerney KA et al. analyzed data from an internet-based prospective cohort study of 2573 female pregnancy planners aged 21–45 years from the USA and Canada. Participants completed a baseline questionnaire and bimonthly follow-up questionnaires until a reported pregnancy or for 12 months, whichever occurred first. They have established that the women who used naproxen or opioids had slightly lower fecundability than women who did not use any pain-relieving medications; use of acetaminophen, aspirin, and ibuprofen was not appreciably associated with fecundability [134].

Levonorgestrel (1.5 mg), a standard drug for emergency contraception (EC), is not effective if administered post-ovulation. A COX2 inhibitor, a 40 mg oral dose of piroxicam as co-treatment with levonorgestrel, improved emergency contraceptive efficacy [135]. Other COX2 inhibitors, such as meloxicam, combined with ulipristal-acetate, disrupted ovulation at peak luteal surge and are a promising candidate for evaluation as a pericoital oral contraceptive [136].

Aspirin exerts its effect primarily by interfering with the biosynthesis of cyclic prostanoids TXA2, prostacyclin, and other prostaglandins. A prospective cohort study of women between 30 and 44 years of age who were trying to conceive naturally investigated the effect of aspirin on ovulation. The use of a therapeutic dose of aspirin in the implantation window was associated with increased fecundability. It has been hypothesized that one of the reasons for higher fecundability in the implantation window is due to aspirin’s vasodilatory and anti-inflammatory properties. These properties are thought to improve implantation rates in women undergoing IVF [137]. Another study showed that the co-administration of low-dose aspirin (81 mg) with tamoxifen significantly improves ovarian response to stimulation, endometrial thickness, and pregnancy rates in anovulatory PCOS patients [138].

Indomethacin has a well-documented effect on ovulation because its impact is more substantial and more predictable than aspirin’s. To avoid emergent oocyte retrieval at midnight, indomethacin was administered (150 mg/day, three times a day) from 2 h after the incorrect hCG and GnRH injection to the night before oocyte pickup. The oocyte retrieval was performed at the originally scheduled time. The results showed that pre-operative ovulation was effectively prevented, and a successful collection of the expected number of oocytes was achieved at 45 h after triggering [139].

According to these studies, the use of NSAIDs is mainly associated with the delay or suppression of the ovulation process, which raises questions about their potential harmful effect on the reproductive system. Since NSAIDs are commonly used as acute pain relievers, their periodic usage in appropriate doses probably does not pose a significant risk for women’s fertility; however, chronic, long-term use of these drugs with uncontrolled doses could interfere with the low-grade inflammatory process during ovulation and may increase the appearance of ovulatory impairments. Based on this, it is essential to emphasize that intermittent use of NSAIDs in acute pain or inflammation still stands; however, it is worth mentioning the moderate use of NSAIDs, especially in women around the time of ovulation who are trying to get pregnant.

**Table 3 ijms-26-11979-t003:** Effects on human ovulation of non-steroidal anti-inflammatory drugs (NSAIDs). COX2: cyclooxygenase-2, PCOS: polycystic ovary syndrome, hCG: human chorionic gonadotrophin, GnRH: gonadotropin-releasing hormone.

NSAIDs	COX2 Selectivity	Involved Patients	Dose	Time Window	Results	References
**Aspirin**	low	858 women with regular menstrual cycles	dose is not precisely determined (325–650 mg)	during periods of preovulatory, periovulatory, implantation, and nonmenstrual bleeding days of the cycle	aspirin use around implantation was associated with increased fecundability	[137]
**Aspirin** **+** **Tamoxifen**	low	188 anovulatory PCOS women	81 mg aspirin + 10 mg tamoxifen twice daily from 3rd day to 7th day of the cycle	during the ovulation induction process	improve ovarian response to stimulation, endometrial thickness, and pregnancy rates in anovulatory PCOS patients	[138]
**Ibuprofen**	moderate	51 women with regular menstrual cycles	3 × 400 mg/day	one cycle per patient	delayed ovulation	[130]
		111 women with regular menstrual cycles	6 × 400 mg within 38–48 h	during the in vitro fertilization process	ibuprofen did not harm ovulation as an analgesic treatment	[131]
		9 women aged over 40 or with poor ovarian response	800 mg, after 15–18 h post-trigger injection	during the in vitro fertilization process	significant changes in the follicular fluid inflammatory milieu	[132]
**Indometacin**	low	1 woman aged 42 with primary infertility (case report)	150 mg/day (three times a day) from 2 h after the incorrect hCG and GnRH injection	during the in vitro fertilization process	prevent and postpone ovulation	[139]
**Naproxen**	low	1763 pregnancy planner women	dose is not precisely determined (880 mg mean monthly dose with the range of 220–13,200 mg)	during the previous 4 weeks before the questionnaire	naproxen was associated with a slight reduction in fecundability	[134]
**Levonorgestrel** **+** **Piroxicam**	moderate	860 women required emergency contraception	1.5 mg levonorgestrel + 40 mg piroxicam	within 72 h of unprotected sexual intercourse	prevented 94.7% of expected pregnancies compared with 63.4% for levonorgestrel plus placebo	[135]
**Ulipristal** **+** **Meloxicam**	high	9 women with regular menstrual cycles	30 mg ulipristal + 30 mg meloxicam	one baseline cycle followed by one treatment cycle	disrupts ovulation at the peak luteal surge	[136]

## 6. Discussion

Animal and human studies showed that ovulation could be influenced by the manipulation of the inflammatory and redox signaling pathways with the use of antioxidants or anti-inflammatory drugs. The rat model could be a promising and suitable model for the study of pharmacological interventions on the ovulatory inflammatory process; however, important considerations are necessary regarding the translational potential.

One of the main differences between the two species is the length of their reproductive cycles. The human menstrual cycle lasts approximately 28 days and includes a luteal phase, during which the corpus luteum remains functionally active. In contrast, the cycle in rodents lasts 4–5 days, and the corpus luteum rapidly regresses unless pregnancy occurs. This difference raises the question of whether corpus luteum function and pathological conditions (e.g., anovulation) are directly comparable between the two species. This may limit the translational reliability of results derived from this model.

Furthermore, rats exhibit polyovulation, whereas humans typically ovulate only one egg. This suggests species-specific differences in dominant follicle selection, gonadotropin sensitivity, and intraovarian signaling. Therefore, conclusions drawn from the process of follicular development in rats should be applied with caution when studying human infertility or ovarian aging. Another difference is the mechanism of endometrial renewal. During menstruation, the endometrial layer is shed under the influence of hormones, and abnormalities in this process play a role in pathological conditions such as endometriosis and problems with egg implantation. Rodents develop their endometrium without menstruation, which limits the relevance of the animal model for modeling human endometrial diseases.

Nevertheless, rats remain a valuable model due to their short hormonal cycle, rapid reproduction, shared hormonal regulation (HPO axis), and similar hormonal functions, such as estrogen, FSH, and LH, to those of humans.

Oxidative stress and antioxidants play complex, often contradictory roles in ovarian function. While antioxidants such as SOD isoforms, GPx, CAT, and non-enzymatic molecules protect follicles from ROS-induced damage, ROS are essential for ovulation. This dual function poses a serious challenge: excessive ROS impairs egg quality and accelerates ovarian aging, while insufficient ROS—due to high antioxidant activity—can inhibit cumulus expansion, steroidogenesis, and follicle rupture. Knowledge of the functioning of the oxidant-antioxidant system during ovulation primarily comes from rodent models, which differ significantly from humans in the ways described above, limiting their direct clinical application.

Literature data support that NSAIDs may influence ovulation through inhibition of COX activity, but these results are contradictory. COX2 clearly plays an important role in periovulatory prostaglandin synthesis, but the clinical effects of NSAIDs on ovulation vary greatly depending on the type of drug, dose, timing, and patient population. Ibuprofen, naproxen, and indomethacin may delay or suppress ovulation. No adverse effects of NSAIDs on egg maturation, embryo quality, or implantation have been found, particularly in an IVF setting. Unfortunately, most studies rely on small sample sizes, short-term exposures, or heterogeneous dosing regimens, making generalization difficult. Furthermore, the combination of COX2 inhibitors and progesterone may be therapeutically beneficial in emergency contraception.

## 7. Conclusions

Humans and rats have similar hormonal changes during the menstrual and estrous cycles, respectively. In both species, LH induction initiates the structural changes of the pre-ovulatory follicle, which results in the release of numerous autocrine and paracrine mediators, including inflammatory and redox signaling molecules. These signaling pathways are interrelated and trigger a self-generating inflammatory and redox cascade process, leading to local inflammation and wall rupture within the follicle.

Lipid-derived mediators are one of the key regulators of the induction of follicular inflammation. All three types of eicosanoids take part in the process of ovulation. Prostaglandins mediate almost all of the vital processes necessary for follicle wall rupture and oocyte release, including the vasculature changes, ECM remodeling and degradation, and COC expansion. However, leukotrienes have similar functions as prostaglandins; their role is mainly limited to the support of leukocyte infiltration into the follicle, while thromboxane seems to be negligible in the initiation of ovulation, but it is presumably significant after the rupture to control the blood loss. In contrast to this, the redox balance is essential both in the follicular maturation and the ovulation process. Excess ROS can lead to impaired oocyte maturation and development, which manifests in fertility impairments; however, excess antioxidants can interfere with ovulation by inhibiting the production of ROS that is necessary for ovulation under physiological conditions.

Both humans and rats exhibit similar characteristics of eicosanoids and enzymatic antioxidants, which are influenced by the LH surge. Currently, studies with the direct effect of antioxidants on the ovulation rate are limited to in vivo animal models and warrant further clinical evidence.

Occasional use of NSAIDs is unlikely to impair human fertility significantly, but chronic or poorly timed use may interfere with the low-grade inflammatory processes necessary for follicular rupture. Current evidence is not entirely clear, so caution is advised regarding regular use of NSAIDs in women who are actively trying to conceive or undergoing assisted reproduction. More large-scale, controlled studies are needed to clarify the exact processes and effects.

## Figures and Tables

**Figure 1 ijms-26-11979-f001:**
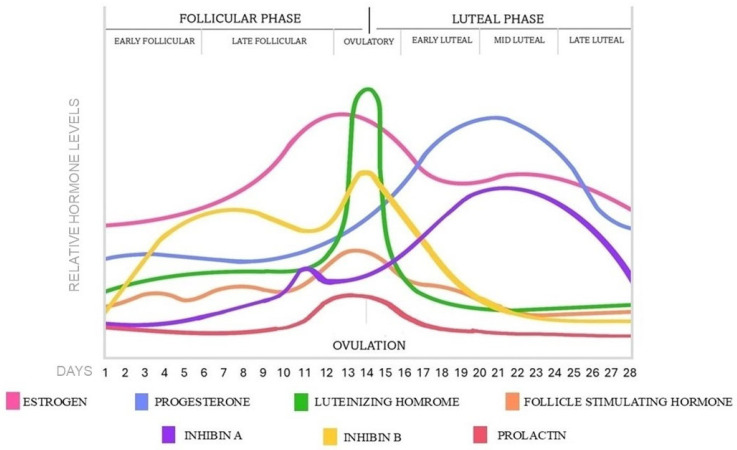
Hormonal changes and phases in a 28-day menstrual cycle.

**Figure 2 ijms-26-11979-f002:**
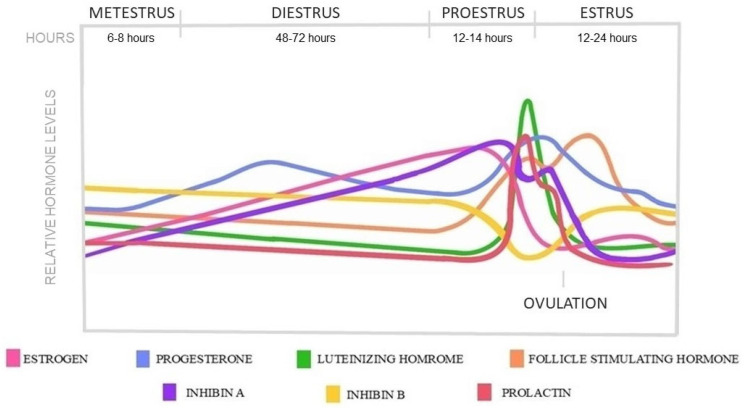
Hormonal changes in a 5-day rat estrus cycle.

**Figure 3 ijms-26-11979-f003:**
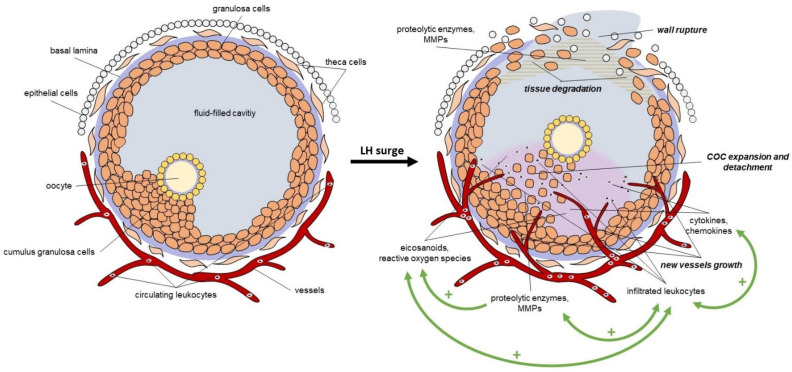
The LH-induced local inflammation in the follicle during ovulation. LH: luteinizing hormone, MMPs: matrix metallopeptidases, COC: cumulus oocyte complex, +: positive feedback loop.

**Table 1 ijms-26-11979-t001:** Comparison of the 28-day human menstrual and 5-day rat estrus cycles.

**Menstrual Cycle**						28 Days (varied between 21–35 days)		
**HUMAN**	**STAGES**		**FOLLICULAR PHASE**		**OVULATION**	**LUTEAL PHASE**		
			Early	Late		Early	Mid	Late
			1–5 days	6–14 days	12–36 h after the LH surge	15–20 days	21–23 days	24–28 days
	**HORMONAL** **PROFILE**	FSH level	↔	↑	pre-ovulatory peak	↓	↔	↔
		LH level	↔	↔	pre-ovulatory peak	↔	↔	↔
		E level	↔	↑	pre-ovulatory peak	moderate ↓	second peak	↓
		P4 level	↔	↔	↔	↑	peak	↓
		INH-A level	↔	↔	pre-ovulatory ↑	↑	peak	↓
		INH-B level	↔	↑	pre-ovulatory peak	↓	↓	↔
		PRL level	↔	↔	slight ↑	slight ↑	peak	↓
	**REPRODUCTIVE EVENT**		menstruation	follicular maturation	follicle rupture, oocyte release (1/cycle)	corpus luteum develop and function		luteolysis

**RAT**	**REPRODUCTIVE EVENT**		follicular maturation		follicle rupture, oocyte released (>1/cycle)	sexual receptivity	corpus luteum develop and regression	luteolysis, endometrium reabsorption
	**HORMONAL** **PROFILE**	PRL level	↑		pre-ovulatory peak	↔	↔	↔
		INH-B level	↔		minimum	↔	↔	↔
		INH-A level	peak		↓	↔	↑	↑
		P4 level	↔		pre-ovulatory peak	↑	↓	↔
		E level	↑		pre-ovulatory peak	↔	↔	↑
		LH level	↔		pre-ovulatory peak	↔	↔	↔
		FSH level	↑		pre-ovulatory peak	↓	↔	↔
	**STAGES**		0–12 h		10–12 h after the LH surge	12–48 h	48–54 h	54–120 h
			**PROESTRUS**		**OVULATION**	**ESTRUS**	**METESTRUS**	**DIESTRUS**
**Estrus cycle**						5 days (varied between 4–5 days)		

FSH: follicle-stimulating hormone, LH: luteinizing hormone, E: estrogen, P4: progesterone, INH-A: inhibin A, INH-B: inhibin B, PRL: prolactin, ↔: baseline level, ↑: level increase, ↓: level decrease.

**Table 2 ijms-26-11979-t002:** Comparison of bioactive agents in human and rat ovaries. PGE2: prostaglandin E2, PGF2α: prostaglandin F2-alfa, SOD: superoxide dismutase, CAT: catalase, GPx: glutathione peroxidase, GST: glutathione transferase, GSH: glutathione, ↑: increase, hCG: human chorionic gonadotrophin, Gn: gonadotropin.

	Human	Rat	References
**Eicosanoids**			
**PGE2**	follicular fluid follicular cells	follicular cells (↑ by Gn)	[53,56,69,106,107,108]
**PGF2α**	follicular fluid (↑ on day 14 of cycle)	follicular cells (↑ by Gn)	[53,56,106,108,109,110,111]
**Thromboxane B2**	follicular fluid	ovarian cells (↑ by Gn)	[53,56,65,66,109,112]
**Leukotriene B4**	follicular fluid	ovarian cells (↑ by Gn)	[56,69,73,106]
**Antioxidants**			
**SOD**	follicular fluid (↑ level than serum after hCG) follicular cells (↑ with maturation)	ovarian cells (↑ by Gn)	[79,81,84,102,103,113,114,115,116]
**CAT**	follicular fluid follicular cells	follicular cells ovarian cells (↑ by Gn)	[81,113,115,117,118,119]
**GPx**	follicular fluid follicular cells	follicular cells ovarian cells	[113,114,115,120,121]
**GST**	follicular fluid follicular cells	ovarian cells (↑ by Gn)	[114,115,119,122,123,124]
**GSH**	follicular fluid	follicular cells ovarian cells (↑ by Gn)	[81,125,126]

## Data Availability

No new data were created or analyzed in this study. Data sharing is not applicable to this article.

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
