# Peer review of "Inflammatory and Redox Mediators in Rat and Human Ovulation"

_ijms, 2025, doi:10.3390/ijms262411979_

Round 1

Reviewer 1 Report

Comments and Suggestions for Authors

The manuscript reviews “Inflammatory and redox mediators in rat and human ovulation”. Overall, the topic holds merit, though it needs some improvements before possible publication in the journal.

  1. The introduction should more sharply define the "double-edged sword" nature of both inflammation (controlled process vs. tissue destruction) and redox signaling (molecular messengers vs. oxidative stress). This establishes the conceptual framework of balance that is critical to understanding ovulation.
  2. The authors should address key questions: When is the rat model most predictive? Where has it been misleading? How do differences in immune cell recruitment or antioxidant capacity between species affect the interpretation of data?
  3. This paragraph (L 29-32) correctly identifies the key mediators but presents them as separate, parallel actors. To strengthen it, explicitly frame them as components of an integrated signaling network from the outset, where lipid mediators (e.g., prostaglandins) and redox signals (e.g., ROS) directly regulate and amplify each other's production and actions to drive the inflammatory ovulatory cascade.
  4. The review does not sufficiently advance the reader's understanding beyond foundational textbook knowledge, failing to synthesize conflicting data, or critically evaluate the limitations of key studies.
  5. The major missing is the specific knowledge gaps, unresolved debates, or limitations that this review intends to address
  6. The manuscript, in its current form, is critically undermined by pervasive weaknesses in language precision, syntactic structure, and logical flow.
  7. Most of the references are outdated. Updating the references will ensure the manuscript remains timely and relevant.

Author Response

Answers to Reviewer #1:

The authors would like to thank Reviewer #1 for the suggestions, which have promoted the creation of a better manuscript concerning its scientific value. Our answers are given below. The changes based on the recommendations of Reviewer #1 are marked in green in the revised manuscript. As some of the recommendations of Reviewer #1 overlapped with Reviewer #3's suggestions, these changes were marked in orange in the revised manuscript.

The manuscript reviews “Inflammatory and redox mediators in rat and human ovulation”. Overall, the topic holds merit, though it needs some improvements before possible publication in the journal.

  1. The introduction should more sharply define the "double-edged sword" nature of both inflammation (controlled process vs. tissue destruction) and redox signaling (molecular messengers vs. oxidative stress). This establishes the conceptual framework of balance that is critical to understanding ovulation.

We thank Reviewer #1 for their suggestion. We supplemented the introduction part on the first page with the “double-edged sword nature” of the inflammation and redox signaling in the context of fertility.

  1. The authors should address key questions: When is the rat model most predictive? Where has it been misleading? How do differences in immune cell recruitment or antioxidant capacity between species affect the interpretation of data?

We supplemented the revised manuscript with a discussion section that addresses the strengths and limitations of using rats in ovulatory investigations. We also highlight the similarities and differences between the two species in the context of ovulation. Lines: 361-382.

  1. This paragraph (L 29-32) correctly identifies the key mediators but presents them as separate, parallel actors. To strengthen it, explicitly frame them as components of an integrated signalling network from the outset, where lipid mediators (e.g., prostaglandins) and redox signals (e.g., ROS) directly regulate and amplify each other's production and actions to drive the inflammatory ovulatory cascade.

We agree with this comment; therefore, as Reviewer #1 suggested, we supplemented this paragraph with information on the integrated signaling network, in which lipid and redox mediators directly regulate each other. Since these cascade reaction processes are very complex and multifactorial, we detailed the selected mediators separately for better understanding. We also wrote it in the revised manuscript. Lines 30-41.

  1. The review does not sufficiently advance the reader's understanding beyond foundational textbook knowledge, failing to synthesize conflicting data or critically evaluate the limitations of key studies.

We believe that the manuscript, as revised with the suggestions from the three reviewers, offers readers more than just basic knowledge. At the start of each section, we provide detailed information on the topic, then discuss the underlying mechanisms, functions, and controversies based on existing literature to give deep scientific insight into each subject.

  1. The major missing is the specific knowledge gaps, unresolved debates, or limitations that this review intends to address

By changing the discussion, we believe we have succeeded in clarifying the limitations of comparing rat and human processes. The detailed literature review of the topic ensures that the current version of the manuscript does not contain specific knowledge gaps. Of course, as with all scientific work, it may raise new questions and unresolved issues.

  1. The manuscript, in its current form, is critically undermined by pervasive weaknesses in language precision, syntactic structure, and logical flow.

The manuscript is checked by a language proofreader and AI-based software to eliminate errors in interpretation and grammar.

  1. Most of the references are outdated. Updating the references will ensure the manuscript remains timely and relevant.

During manuscript preparation, we paid special attention to incorporating more recent and comprehensive studies to improve the timeliness and relevance of the manuscript. Our revised manuscript contains 139 references with the following distribution of publication time: 37% of publications are from 2020 or later, 30% of publications are from between 2010 and 2020, 9% of publications are from between 2000 and 2010, 13% of publications are from between 1990 and 2000, and 11% of publications are from between 1976 and 1990. The original research examining the inflammatory characteristics of ovulation was conducted in the 1980s-90s. We included these references in the manuscript because they provide the basic experimental information in the field and ensure a comprehensive assessment of the topic. Regardless, 67% of the literature utilized for the manuscript originated from the last 15 years.

Reviewer 2 Report

Comments and Suggestions for Authors

Review Report

concerning the manuscript submitted to Int. J. Mol. Sci, Manuscript ID: ijms-4002143

by Dorottya Varga et al.

General: In my opinion the Authors present a systematic review of the female ovarian and estrous cycles in the rat and human, describing the LH-inflammatory/redox cascade, the role of eicosanoids (especially prostaglandins) and ROS/antioxidants, and concluding with a discussion of the effects of NSAIDs on ovulation. The work includes useful graphics (Figs. 1–3) and comparative tables (Tables 1–3). The scope of the topic and the selection of references are ambitious (many citations from 2023 to 2025). The manuscript requires some substantive and linguistic/editorial improvements: there are factual inaccuracies (including incorrect abbreviation expansions), minor contradictions, and several unfortunate wording errors.

Major comments:
  1. The section on prostaglandins states: "PTGS1 (COX1) described as cyclooxygenase-2 and PTGS2 (COX2)" – this is an error. PTGS1 ≡ COX-1, PTGS2 ≡ COX-2. Please correct the text and consistently use the correct pairs (PTGS1/COX-1, PTGS2/COX-2) and check whether further conclusions are based on this error (p. 8, line 54).
  2. On p. 5, it is written that "ovulation halves the 28-day menstrual cycle, ovulation occurs at the beginning of the cycle." This is incorrect – ovulation is pericentral (around day 14 of the 28-day cycle). Please correct it and ensure that the diagram and description are consistent with this fact.
  3. The conclusion suggests that “excessive” antioxidant intake may interfere with ovulation; however, the text is dominated by animal models and pharmacological interventions (e.g., SOD, exogenous antioxidants) – please clearly separate animal from clinical evidence and strengthen the limitations section (pp. 12–15, 15).
  4. The title "NSAIDs and influencing of ovulation" is ambiguous – I suggest "NSAIDs and their influence on ovulation."
  5. In Table 3, there is no dose ("no data") for naproxen, and different schedules for ibuprofen – it would be worthwhile to standardize the presentation (doses, time window, population) and add it.
  6. Please verify the PGH₂ transformation pathways (the role of aldo-keto reductase isoenzymes) and EP/FP receptor records to avoid overgeneralizations.
  7. Since this is a review paper, it is worth adding a short methods section (databases, date range, keywords, inclusion/exclusion criteria) to enhance reliability. Please add this section at the end of the Introduction section.

Minor comments:
  1. Please standardize the English: "estrus" vs. "oestrus" (Table 1 uses "Oestrus," Figure 2 – "estrus").
  2. Introduction: "antioxidant clays (nutritional supplements, vitamins)" – is this probably referring to antioxidants? Please correct (p. 1).
  3. Consistently use the abbreviation COC and the full term ("cumulus-oocyte-complex" vs "cumulus oocyte complex").
  4. Figure 1 and Figure 2: Please label the axes: x-axis: days/hours; y-axis: relative level. Please correct the description "28-days" → "28-day."
  5. I really like Figure 3 and find it very helpful. However, please add positive feedback loop arrows and a legend for proteases/MMPs.
  6. I also highly value Table 1; it's excellent, but minor corrections to the nomenclature are needed ("Estrous" vs. "Oestrus," please standardize).
  7. Please standardize the DOI format and capitalization in titles; check for typos and duplicates (e.g., [56,56]).
  8. Some references are not written according to mdpi editorial requirements, please correct them (e.g. ref. 35).
  9. Please ensure that all abbreviations in the text are defined upon first use (e.g., Prdx, Nrf2) (pp. 12–16, 18–23).
Considering that the manuscript requires few revisions, I believe that after incorporating all my comments and suggestions, it can be published, subject to approval by the Editorial Board. Recommendation: Minor Revision.

Author Response

Answers to Reviewer #2:

The authors would like to thank Reviewer #2 for the suggestions, which have promoted the creation of a better manuscript concerning its scientific value. Our answers are given below. Changes are marked in purple in the manuscript.

Comments:

In my opinion the Authors present a systematic review of the female ovarian and estrous cycles in the rat and human, describing the LH-inflammatory/redox cascade, the role of eicosanoids (especially prostaglandins) and ROS/antioxidants, and concluding with a discussion of the effects of NSAIDs on ovulation. The work includes useful graphics (Figs. 1–3) and comparative tables (Tables 1–3). The scope of the topic and the selection of references are ambitious (many citations from 2023 to 2025). The manuscript requires some substantive and linguistic/editorial improvements: there are factual inaccuracies (including incorrect abbreviation expansions), minor contradictions, and several unfortunate wording errors.

  1. The section on prostaglandins states: "PTGS1 (COX1) described as cyclooxygenase-2 and PTGS2 (COX2)" – this is an error. PTGS1 ≡ COX-1, PTGS2 ≡ COX-2. Please correct the text and consistently use the correct pairs (PTGS1/COX-1, PTGS2/COX-2) and check whether further conclusions are based on this error (p. 8, line 54).

We thank Reviewer #2 for their suggestion. We reviewed and corrected the text according to the correct enzyme pairs and other related errors. Lines: 62-66.

  1. On p. 5, it is written that "ovulation halves the 28-day menstrual cycle, and ovulation occurs at the beginning of the cycle." This is incorrect – ovulation is pericentral (around day 14 of the 28-day cycle). Please correct it and ensure that the diagram and description are consistent with this fact.

We agree with this comment; therefore, as Reviewer #2 suggested, we corrected this in the revised manuscript as follows on page 5: “Since in humans, the ovulation is pericentral, it halves the 28-day menstrual cycle and occurs around day 14 of the 28-day cycle; however, each phase has its own counterpart.” This statement aligns with the diagram, table, and descriptions.

  1. The conclusion suggests that “excessive” antioxidant intake may interfere with ovulation; however, the text is dominated by animal models and pharmacological interventions (e.g., SOD, exogenous antioxidants) – please clearly separate animal from clinical evidence and strengthen the limitations section (pp. 12–15, 15).

The conclusion part has been rewritten, focusing on the separation of animal data and clinical evidence as Reviewer #2 recommended. Lines: 383-408.

  1. The title "NSAIDs and influencing of ovulation" is ambiguous – I suggest "NSAIDs and their influence on ovulation."

Thanks for the suggestion of Reviewer #2. We retitled section 4, as suggested in line 280.

  1. In Table 3, there is no dose ("no data") for naproxen, and different schedules for ibuprofen – it would be worthwhile to standardize the presentation (doses, time window, population) and add it.

We supplemented Table 3 with the doses of naproxen, the detailed patient population, and the time windows of each study.

  1. Please verify the PGH₂ transformation pathways (the role of aldo-keto reductase isoenzymes) and EP/FP receptor records to avoid overgeneralizations.

As Reviewer #2 suggested, we verified the role of aldo-keto reductase isoenzymes in prostaglandin transformations and supplemented the text with EP/FP receptor records. Lines: 64-66, 72-74.

  1. Since this is a review paper, it is worth adding a short methods section (databases, date range, keywords, inclusion/exclusion criteria) to enhance reliability. Please add this section at the end of the Introduction section.

Thank you for the helpful suggestion of Reviewer #2; however, we would like to clarify that the submitted manuscript is not a systematic review. Our manuscript is a narrative literature review; therefore, the purpose was to summarize and provide a comprehensive and critical insight into the given topic based on the relevant literature data. Since narrative literature reviews have a more flexible and less formal methodology, rigid methodology elements were not applied.

Minor comments:

  1. Please standardize the English: "estrus" vs. "oestrus" (Table 1 uses "Oestrus," Figure 2 – "estrus").

We standardized the wording in Table 1, as Reviewer #2 suggested.

2. Introduction: "antioxidant clays (nutritional supplements, vitamins)" – is this probably referring to antioxidants? Please correct (p. 1).

We corrected the wording on page 1 as recommended.

3. Consistently use the abbreviation COC and the full term ("cumulus-oocyte-complex" vs "cumulus oocyte complex").

We standardized the full term word of COC in the abbreviation list.

4. Figure 1 and Figure 2: Please label the axes: x-axis: days/hours; y-axis: relative level. Please correct the description "28-days" → "28-day."

We labeled the x and y axes on Figures 1 and 2 according to the recommendations of Reviewer #2. We also corrected the description of Figure 1 as suggested.

5. I really like Figure 3 and find it very helpful. However, please add positive feedback loop arrows and a legend for proteases/MMPs.

We added positive feedback loop arrows and legends for proteases/MMPs on Figure 3.

6. I also highly value Table 1; it's excellent, but minor corrections to the nomenclature are needed ("Estrous" vs. "Oestrus," please standardize).

As previously described, we standardized the wording in Table 1 according to the estrus cycle.

7. Please standardize the DOI format and capitalization in titles; check for typos and duplicates (e.g., [56,56]).

We corrected the reference duplicates in the revised manuscript in line 125. The citations and reference list were created by using Zotero, and the reference style of the manuscript follows the journal's guidelines and requirements. We checked the formats and corrected where necessary.

8. Some references are not written according to MDPI editorial requirements; please correct them (e.g., ref. 35).

The citations and reference list were created by using Zotero, and the reference style of the manuscript follows the journal's guidelines and requirements. We corrected the formats where necessary, involving reference 35 as suggested.

9. Please ensure that all abbreviations in the text are defined upon first use (e.g., Prdx, Nrf2) (pp. 12–16, 18–23).

We checked and corrected the abbreviations in the revised manuscript.

Reviewer 3 Report

Comments and Suggestions for Authors

I have read the manuscript “Inflammatory and redox mediators in rat and human ovulation” written by Dorottya Varga, Péter Szatmári and Eszter Ducza and have several concerns.

1.      Please check the author's affiliation. Seems like “21” is a typo; check, please. Please verify the whole text for other minor grammatical and typo mistakes.

2.      The manuscript would benefit from the inclusion of a discussion section that addresses the limitations of the study, outlines future perspectives, and explains the necessity of further investigation into this topic. Additionally, it should provide information about inhibitors of IL-1, IL-6, and TNF. Moreover, it seems to be interesting to discuss pros and cons in the usage of NSAIDs and other anti-inflammatory drugs from the point of view of the reproductive system.

3.      The review would benefit from further exploration of what constitutes “physiologically coordinated” inflammation and redox state. What distinguishes this from pathological inflammation that leads to infertility? Additionally, could stress be more significant than inflammation in this context? What is the difference between it and pathological inflammation that leads to infertility? Could stress be more important than inflammation?

4.      Where is the line between positive and negative impacts of inflammation?

5.      While obtaining an animal model for studying the ovulatory processes is important, it is more important to understand its limits. This review would benefit from a discussion on the limitations of the rat model. Additionally, an important consideration is how to translate findings from rat studies to human applications. What specific experimental approaches or biomarker studies in humans are needed to confirm these findings directly?

6.      Please specify which NSAID inhibits which specific COX isoforms (COX-1 vs. COX-2) in the context of ovulation.

Author Response

Answers to Reviewer #3:

The authors would like to thank Reviewer #3 for the suggestions, which have promoted the creation of a better manuscript concerning its scientific value. Our answers are given below. The changes based on the recommendations of Reviewer #3 are marked in blue in the revised manuscript. As some of the recommendations of Reviewer #3 overlapped with some suggestions of Reviewer #1, these changes were marked with orange colour in the revised manuscript.

Comments:

I have read the manuscript “Inflammatory and redox mediators in rat and human ovulation” written by Dorottya Varga, Péter Szatmári and Eszter Ducza and have several concerns.

  1. Please check the author's affiliation. Seems like “21” is a typo; check, please. Please verify the whole text for other minor grammatical and typo mistakes.

We thank Reviewer #3 for this observation. We corrected this typo in the revised manuscript and checked the whole text for other mistakes.

  1. The manuscript would benefit from the inclusion of a discussion section that addresses the limitations of the study, outlines future perspectives, and explains the necessity of further investigation into this topic. Additionally, it should provide information about inhibitors of IL-1, IL-6, and TNF. Moreover, it seems to be interesting to discuss pros and cons in the usage of NSAIDs and other anti-inflammatory drugs from the point of view of the reproductive system.

We thank Reviewer #3 for their suggestion. We supplemented the revised manuscript with a discussion section addressing limitations, future perspectives, and investigations. Lines: 361-382. This review aimed to summarize the LH-induced inflammatory and redox cascade involved in follicle rupture through the details of lipid-derived and redox signaling mediators. Although these signaling mediators are strongly interconnected with cytokines, chemokines, and the immune system as well, we did not present them in detail. A more detailed discussion of these processes, including cytokines, chemokines, and the immune system, would involve a huge level of information that would exceed the specified scope of the manuscript, and the focus on eicosanoids and redox mediators would be lost. We believe that the discussion of these processes, including cytokines, chemokines, and the immune system, should be summarized in another manuscript.

As recommended, we discussed the use of NSAIDs in the revised manuscript, lines 351-360. Other anti-inflammatory drugs were not discussed, since the manuscript details only the effects of NSAIDs on ovulation.

  1. The review would benefit from further exploration of what constitutes “physiologically coordinated” inflammation and redox state. What distinguishes this from pathological inflammation that leads to infertility? Additionally, could stress be more significant than inflammation in this context? What is the difference between it and pathological inflammation that leads to infertility? Could stress be more important than inflammation?

We thank Reviewer #3 for their suggestion. We supplemented the introduction part on page 1 with the “double-edged sword nature” of the inflammation and redox signaling in the context of fertility. Since the role of oxidative stress and inflammation in infertility has been extensively reviewed in recent years, our review focused on the physiological inflammation and redox reaction. Therefore, we did not discuss these relations in detail; however, we addressed these processes to give a frame to our manuscript.

  1. Where is the line between positive and negative impacts of inflammation?

Following the Reviewer #3 recommendation, the introduction part was supplemented with extra information on the positive and negative impacts of inflammation on fertility. It is hard to determine where the specific line is between positive and negative impacts of inflammation; however, the negative effect of the inflammation (which is usually associated with tissue degradation) positively correlates with the duration and extension of the inflammatory event. We mentioned this in the revised introduction part in the context of infertility.

  1. While obtaining an animal model for studying the ovulatory processes is important, it is more important to understand its limits. This review would benefit from a discussion on the limitations of the rat model. Additionally, an important consideration is how to translate findings from rat studies to human applications. What specific experimental approaches or biomarker studies in humans are needed to confirm these findings directly?

We thank Reviewer #3 for their suggestion. We supplemented the revised manuscript with a discussion section addressing limitations and future perspectives. Furthermore, we also highlighted the specific experimental approaches that are promising in the translational process from rat to human. Lines: 361-382.

  1. Please specify which NSAID inhibits which specific COX isoforms (COX-1 vs. COX-2) in the context of ovulation.

Thanks for the helpful comments of Reviewer #3. Following the recommendation, we supplemented the revised manuscript with information on the COX specificity of NSAIDs in lines 289-291 and in Table 3.

Round 2

Reviewer 1 Report

Comments and Suggestions for Authors

None

Author Response

The authors thank the reviewer for her/his work and valuable suggestions, which improved the manuscript.